# Mathematically modeling spillovers of an emerging infectious zoonosis with an intermediate host

Katherine Royce◉*, Feng Fu◉

Dartmouth College Mathematics Department, Hanover, NH, United States of America

◉ These authors contributed equally to this work.
* katherine.p.royce.19@dartmouth.edu

**Data Availability Statement:** All relevant data are within the manuscript and its Supporting Information files.

**Funding:** The author(s) received no specific funding for this work.

## Abstract

Modeling the behavior of zoonotic pandemic threats is a key component of their control. Many emerging zoonoses, such as SARS, Nipah, and Hendra, mutated from their wild type while circulating in an intermediate host population, usually a domestic species, to become more transmissible among humans, and this transmission route will only become more likely as agriculture and trade intensifies around the world. Passage through an intermediate host enables many otherwise rare diseases to become better adapted to humans, and so understanding this process with accurate mathematical models is necessary to prevent epidemics of emerging zoonoses, guide policy interventions in public health, and predict the behavior of an epidemic. In this paper, we account for a zoonotic disease mutating in an intermediate host by introducing a new mathematical model for disease transmission among three species. We present a model of these disease dynamics, including the equilibria of the system and the basic reproductive number of the pathogen, finding that in the presence of biologically realistic interspecies transmission parameters, a zoonotic disease with the capacity to mutate in an intermediate host population can establish itself in humans even if its $R_0$ in humans is less than 1. This result and model can be used to predict the behavior of any zoonosis with an intermediate host and assist efforts to protect public health.

## Introduction

Zoonotic diseases, which originate in animals and infect humans, are one of the most concerning epidemic threats of the 21$^{st}$ century and form 60% of all known infectious diseases [1]. Zoonoses such as HIV-AIDS, avian influenza, SARS, Ebola, Nipah, Hendra, and rabies all trace their origin to nonhuman reservoir species [2], and zoonoses comprise 75% of emerging infectious disease [3]. The World Health Organization cites "Disease X", a pathogen currently unknown to cause human disease that might evolve to become more transmissible among humans, as a priority for research and development in pandemic prevention [4], a threat underscored in recent months by the SARS-CoV2 pandemic.

**Competing interests:** The authors have declared that no competing interests exist.

While the dynamics of a zoonosis in its reservoir host are frequently cited as an influence on its emergence in humans [1], current mathematical models of zoonoses lack the capacity to represent their complete evolution. Some of the most pressing unaddressed questions in establishing the mathematical theory of zoonoses include better capturing disease dynamics within nonhuman species in order to characterize changes in the disease before it infects humans; focusing on the first cases of human infection to understand how a pathogen actively adapts to humans; and developing a theory for the role of intermediate hosts in the emergence of the disease [5]. Lloyd et al. (2009) [6] blame a desire to view zoonoses in a piecewise manner, as a concatenation of different epidemics rather than a connected system, for the lack of quantitative understanding of zoonoses as a new type of disease; in particular, there are few unifying mathematical theories or sets of principles that can be used to frame discussions of zoonotic spillovers [5]. This gap in modeling spillover dynamics limits our understanding of zoonoses, as does a general lack of mathematical modeling of multihost pathogens and quantification of the rate of human-to-human transmission [6, 7]. This paper provides such a mathematical model for a zoonosis emerging through an intermediate host.

In contrast to pathogens which evolved to infect humans, such as smallpox, the biology of emerging zoonoses is adapted to their reservoir host species. Since a pathogen's transmissibility can also be affected by anthropogenic factors such as the host species' population structure or resource and habitat availability [7], intermediate hosts—a non-reservoir animal species in which a zoonotic pathogen circulates—particularly domestic animals, provide greater opportunity for a pathogen to mutate to a human-transmissible form, because these species are biologically similar to the pathogen's wild reservoir and have greater contact with humans. As an example of the role of intermediate hosts, the adaptation of avian influenza, one of the most well-studied zoonoses, to humans requires a mutation in domestic pigs or poultry. Avian influenza's success in a new host species is governed by its receptor binding specificity [8]; with circulation in domestic pigs, which express both human- and avian-influenza type receptors in their tracheae, the virus has an opportunity to mutate to a form that can infect humans ([9], [10]). The influenzas are perhaps the easiest example to understand, as reassortment of different hemagglutinin and neuraminidase subtypes within one infected pig can produce entirely new pathogens [9], but less drastic mutations can alter the transmissibility or lethality of any zoonosis. The disease dynamics that resulted from repeated introductions of Nipah virus from bats, the pathogen's reservoir host, to pigs enambled the pathogen to persist in its intermediate host and thus infect humans [11–13]). Table 1, a sampling of zoonoses for which an intermediate host has been identified, shows notable case studies of zoonoses with domesticated species as intermediate hosts.

**Table 1. Zoonotic diseases with intermediate hosts.**

| Disease | Reservoir Host | Intermediate Host | Source |
|---|---|---|---|
| Nipah virus encephalitis | bats | pigs | [1, 2, 5, 11] |
| Hendra virus disease | bats | horses | [2, 5, 11] |
| SARS | bats | civets | [5] |
| Avian influenza | wild birds | domestic poultry, pigs | [16, 20, 21] |
| Menangle virus disease | bats | pigs | [2, 11] |
| Middle East Respiratory Syndrome | bats | camels | [22] |
| Campylobacteriosis | wild birds | domestic poultry | [23] |
| Japanese encephalitis | wild birds | pigs | [23] |
| Covid-19 | bats | unknown | [24] |

In an intermediate host species, a pathogen can gain more exposure to humans and mutate to a human-transmissible form, an evolution not previously studied. Childs et al. (2019) [14] consider the risk of yellow fever spillover in Brazil, but do not investigate reservoir infection dynamics nor consider pathogen mutation over the course of an epidemic. Similarly, Washburne et al. (2019) [15] introduce percolation models of pathogen spillover in an attempt to capture the complexity of multispecies diseases, but note that this type of model does not capture epidemiological feedback between nonhuman species. Iwami et al. (2007) [16] and Gumel et al. (2009) [17] conceptualize avian influenza mutation as occurring within humans rather than another species, a framework which ignores the key population in the spread of a zoonosis: Richard et al. (2014) [8] cite two barriers, jumping to humans and efficient human-to-human transmission, that a zoonotic pathogen must overcome, and this change frequently occurs in the "mixing vessel" of an intermediate host species [9]. Plowright et al. (2017) [18] present a conceptual model of spillover intended to assess zoonotic risk and identify barriers to spillover, but their quantitative model lacks SIR dynamics in nonhuman species, instead conceptualizing disease in animals as merely a force of infection applied to the human population, and makes no mention of the crucial role played by intermediate hosts. Further, controlling a human epidemic of a zoonotic disease depends on controlling the basic reproduction number in both animals and humans [19], interventions not previously studied together. With a mathematical theory for a human-transmissible disease arising from a zoonotic pathogen in an intermediate host population, researchers can investigate the cumulative effect of evolution in multiple species and policymakers can move towards prevention of a human pandemic rather than amelioration of one [5]. While recent modeling efforts have addressed the spillover process from reservoir host to humans, the role of intermediate hosts as amplifiers or mutators of a pathogen, a defining part of zoonotic spillover, remains underdeveloped and lacks a strong theoretical foundation.

The model presented here is based on the basic SIR model first presented by Kermack and McKendrick (1927) [25], as well as the introduction to multihost SIR models presented by Allen et al. (2012) [7]. We build on more well-known examples such as models for vector-borne diseases, which must infect both its host species (rather than opportunistically jumping to a new species) and follows set steps in its life cycle in both (rather than unpredictably mutating in a new host), contrasting our model with a vector-borne SIR one which merely adds more compartments for the pathogen to run through. Andraud et al. (2012) [26], in a review paper of deterministic models of dengue, note that the disease dynamics among the vector population are frequently simplified to a mere force of infection for the human one, since the disease does not evolve within the vector species. In contrast, a zoonosis model must consider the disease dynamics in its nonhuman compartments, since these dynamics determine whether the pathogen reaches humans at all. Attempts have been made to model zoonotic spillovers [6, 7, 27], but without incorporating changes in the pathogen's ecology over the course of an epidemic, these models are mathematically indistinguishable from those modeling a vector-borne disease with more hosts or a multispecies model. While a sizeable literature exists on mathematical models of vectorborne diseases, and this class of pathogen provides a useful comparison for the type of behavior modeled here, no model captures the unintentional opportunism of zoonoses or incorporates selective pressure on viruses [7]. In this paper, we present a model which incorporates a pathogen mutation to a human-transmissible form in an intermediate host species, filling the gap noted by Lloyd et al. (2015) [5] with the introduction of a mathematical model that simulates the entire course of an emerging zoonosis. We model the adaptation of a zoonotic pathogen to a human-transmissible form in an intermediate host population and investigate whether the presence of pathogen adaptation in intermediate hosts creates or amplifies an epidemic among humans, with the goal of informing public health

efforts to curb emerging infectious diseases. As a baseline and example, we use parameters that most closely reflect highly pathogenic avian influenza, a classical example of a zoonosis with an intermediate host [2] and one for which the most data is available. However, our model is intended to codify the idea of an intermediate host mathematically and therefore does not focus on a particular infectious disease. By changing its parameters, this model can be applied to study any zoonosis that passes through an intermediate host population, and its results are general to that theory.

We find that completely accounting for the spillover and interpopulation dynamics exhibited by emerging zoonoses links human populations to animal ones more deeply than previously thought. Zoonotic diseases are currently classified on the basis of their human-to-human transmissibility [6], which is assumed to be a critical distinction between pathogens with pandemic potential and pathogens that remain relatively rare [1, 3, 5]. The major distinction in zoonotic spread within humans is whether the pathogen can spread beyond its primary individual host to infect other humans: whether the basic reproduction number $R_0$, the number of secondary cases produced by an index case in an entirely naive population, is greater than 1 [5]. This classification rests on a three-stage framework summarized by Lloyd et al. (2009), Morse et al. (2012), and Wolfe et al. (2005) [6, 28, 29]. Stage 1, pre-emergence, represents zoonoses circulating in an intermediate host but only capable of spillover into a dead-end human host, with no further transmission. Stage 2, localized emergence, defines diseases that can maintain stuttering chains in a human population with reinfection from animal hosts but are incapable of sustaining themselves in humans alone. Stage 3, pandemic emergence, classifies diseases that are fully adapted to humans and thus capable of causing outbreaks in our species alone [6, 28].

Here, we examine the process of pathogen evolution through these different stages to show that with a mutation to a human-transmissible strain in an intermediate host, a pathogen can maintain an endemic equilibrium in humans even in stage 1 (an $R_0 < 1$ in the human compartment), refuting the transmissibility framework that currently forms the basis for classification of emerging zoonoses [6, 28, 29]. Since the epidemiological stratification of zoonotic diseases currently rests on their perceived threat to humans, the result that zoonotic epidemics can persist in human populations without achieving an $R_0 > 1$ in humans sounds an alarm for current public health policy.

## Methods

We link three species–a wild reservoir host, a domestic intermediate host, and humans–using a deterministic SIR model [25, 30, 31] with vital dynamics in each species compartment. These compartments are linked by transmission routes. An infected wild host can pass the disease to a susceptible domestic animal with transmission probability $p_d$, and an infected domestic animal can pass the human-transmissible strain of the disease to a human with probability $p_h$. Finally, the model incorporates the hallmark of an emerging zoonosis: the pathogen's ability to mutate to a human-transmissible strain while circulating in a domestic host. To model this phenomenon, we introduce a category $T$ (transmissible) for domestic animals in which the zoonosis has mutated to a human-transmissible form. This mutation happens at a rate $\mu$ in infected domestic animals, who then transition from the original infected category to the transmissible one and can infect other susceptible domestic animals with the new, human-transmissible strain. The full system of 10 ordinary differential equations is shown in Table 2, with subscripts indicating the species (wild, domestic, or human) to which the compartment belongs. Fig 1 provides a representation of the connections between populations, and Table 3 gives the definition of each variable.

**Table 2. ODE systems of our model with three host compartments (species), composed of wild reservoir hosts, intermediate domestic animal hosts, and human hosts.**

| Wild | |
|---|---|
| | $dS_w/dt = b_w - \beta_w S_w I_w - m_w S_w$ |
| | $dI_w/dt = \beta_w S_w I_w - \gamma_w I_w - m_w I_w$ |
| | $dR_w/dt = \gamma_w I_w - m_w R_w$ |
| Domestic | |
| | $dS_d/dt = b_d - \beta_d S_d I_d - p_d S_d I_w - \beta_d S_d T_d - m_d S_d$ |
| | $dI_d/dt = \beta_d S_d I_d + p_d S_d I_w - \mu I_d - \gamma_d I_d - m_d I_d$ |
| | $dT_d/dt = \mu I_d + \beta_d S_d T_d - \gamma_d T_d - m_d T_d$ |
| | $dR_d/dt = \gamma_d I_d + \gamma_d T_d - m_d R_d$ |
| Humans | |
| | $dS_h/dt = b_h - \beta_h S_h I_h - p_h S_h T_d - m_h S_h$ |
| | $dI_h/dt = \beta_h S_h I_h + p_h S_h T_d - \gamma_h I_h - m_h I_h$ |
| | $dR_h/dt = \gamma_h I_h - m_h R_h$ |

As this is an introductory model, we make several assumptions to clarify the essential dynamics of the system. Firstly, we equate the domestic animal recovery and transmission rates for both strains of the pathogen; the human-transmissible strain is different from the wild one only in that its transmission rate in humans is nonzero. We further assume that the population of each compartment is constant over the course of the simulation, with each species' vital dynamics set at replacement rates, and thus calculate the proportion of susceptible, infected, and recovered animals in each species rather than the raw numbers present in each category. To maintain a focus on population biology and the potential for the spread of disease from infected individuals, we do not consider disease-induced mortality; our model is thus best suited to the first phase of diseases such as the 2009 H1N1 pandemic influenza, which spread between hosts in days but can take weeks to kill. Finally, only domestic animals infected with the $T$ strain can pass the disease to humans, although both strains circulate in the

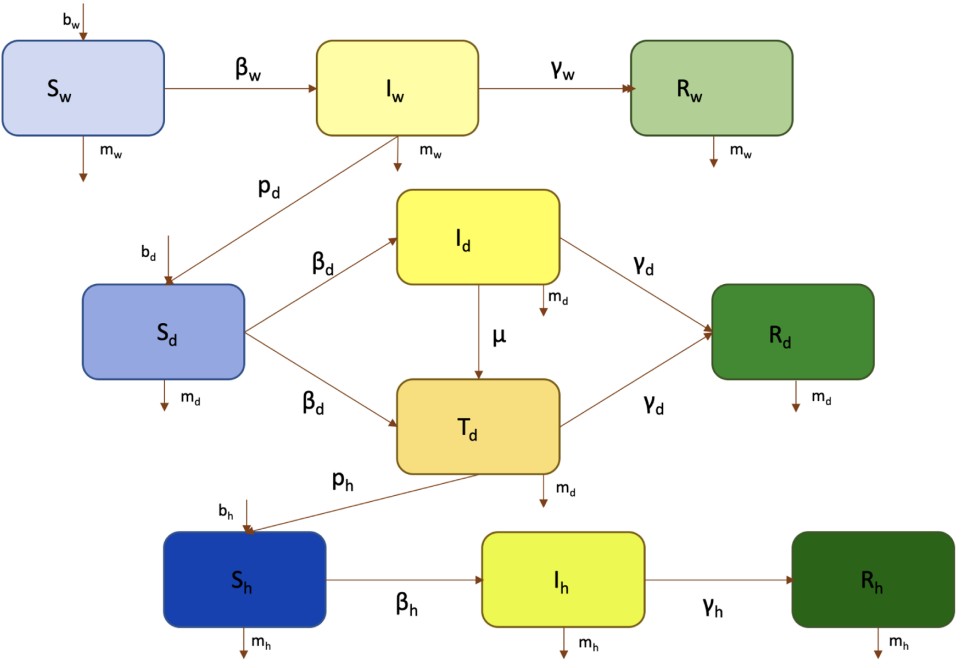

**Fig 1. A representation of the model.** Model parameters are summarized in Table 3.

**Table 3. Parameter definitions.**

| | |
|---|---|
| $S_i$ | susceptible individuals of species $i$ |
| $I_i$ | infected individuals of species $i$ |
| $T_d$ | intermediate hosts infected with human-transmissible strain |
| $R_i$ | recovered individuals of species $i$ |
| $\beta_i$ | transmission rate among species $i$ |
| $\gamma_i$ | recovery rate among species $i$ |
| $b_i$ | birth rate among species $i$ |
| $m_i$ | natural mortality rate among species $i$ |
| $p_d$ | transmission rate from reservoir to intermediate hosts |
| $p_h$ | transmission rate from intermediate hosts to humans |
| $\mu$ | mutation rate of the pathogen in the intermediate host population |

domestic population. The model does not account for coinfection in a domestic animal, since an individual infected with both strains is still capable of starting a human epidemic and is thus counted in the $T$ category.

For each species, the model's value at equilibrium is given by at most a quadratic equation, giving two possible equilibria in each compartment. At the disease-free equilibrium $E_f$, we have $S_i = \frac{b_i}{m_i}$ for each species $i$, while the endemic equilibrium $E_e$ can be shown to satisfy the values shown in Table 4.

We use the next-generation method ([32] and [33]) to calculate $R_0$ in a naive population, giving

$$R_0 = \max\left\{ \frac{\beta_w b_w}{m_w(\gamma_w + m_w)}, \frac{\beta_d b_d}{m_d(\gamma_d + m_d)}, \frac{\beta_d b_d}{m_d(\mu + \gamma_d + m_d)}, \frac{\beta_h b_h}{m_h(\gamma_h + m_h)} \right\}.$$

Note that this approach also gives a distinct reproduction number for each strain in each species: we can define the original pathogen's reproduction number as $R_0^w = \frac{\beta_w b_w}{m_w(\gamma_w + m_w)}$ in the wild compartment and $R_0^d = \frac{\beta_d b_d}{m_d(\gamma_d + m_d)}$, while that of the mutated strain is $R_0^{dm} = \frac{\beta_d b_d}{m_d(\mu + \gamma_d + m_d)}$ in

**Table 4. The endemic equilibria values in each species compartment.** A proof of the uniqueness of the equilibrium value $S_d^*$ is in S1 Appendix.

| | |
|---|---|
| Wild | $S_w^* = \frac{m_w + \gamma_w}{\beta_w}$ |
| | $I_w^* = \frac{b_w - m_w S_w^*}{\beta_w S_w^*}$ |
| | $R_w^* = \frac{\gamma_w I_w^*}{m_w}$ |
| Domestic | $S_d^* < \min\{b_d/(m_d + p_d I_w^*), (\gamma_d + m_d)/\beta_d\}$ |
| | $I_d^* = \frac{1}{\mu}(\gamma_d + m_d - \beta_d S_d^*)T_d^*$ |
| | $T_d^* = \frac{b_d - m_d S_d^*}{(\gamma_d + m_d) + \frac{1}{\mu}(\gamma_d + m_d)(\gamma_d + m_d - \beta_d S_d^*)}$ |
| | $dR_d^* = \frac{\gamma_d(I_d^* + T_d^*)}{m_d}$ |
| Humans | $S_h^* = \frac{\beta_h b_h + (m_h + \gamma_h)(p_h T_d^* + m_d) - \sqrt{[\beta_h b_h + (m_h + \gamma_h)(p_h T_d^* + m_d)]^2 - 4\beta_h m_h b_h(\gamma_h + m_h)}}{2\beta_h m_h}$ |
| | $I_h^* = \frac{b_h - m_h S_h^*}{\gamma_h + m_h}$ |
| | $R_h^* = \frac{\gamma_h I_h^*}{m_h}$ |

the domestic compartment and $R_0^h = \frac{\beta_h b_h}{m_h(\gamma_h + m_h)}$ in humans. While a full analysis of the global stability of the endemic equilibrium requires the Routh-Hurwitz criteria applied to $J(E_e)$, as well as Lyapunov functions specific to the 10-equation system in Table 2 [34], analyzing the eigenvalues of $J(E_e)$ and $J(E_f)$ give the local asymptotic stability for particular parameter values at those equilibria, and we have included examples below. We further note that while $R_0$ retains its traditional value as a threshold for the stability of the disease-free equilibria, it is possible for the disease to vanish from upstream compartments while reaching an endemic equilibria in downstream ones. This behavior is a result of the intercompartment parameters $p_d$, $p_h$, and $\mu$: since the model presented here is deterministic, any positive number of infections in wild animals seeds infections in domestic ones, which in turn transmit the pathogen to humans. Once established in all three species, the fate of the disease in each compartment depends on that species' $R_0^i$. (It is thus possible that the wild species does not serve as a true 'reservoir' host, in which the pathogen perpetually circulates.) However, $R_0$ as defined above measures the stability of the epidemic when considered as a multispecies disease. The model's key innovations are linking three species together based on their proximity to humans and distinguishing between human-transmissible and non-human-transmissible strains of the pathogen, as no previous models simulate either intermediate hosts for zoonoses or a mutation to a human-transmissible form during the course of the epidemic in animals to study the entire range of an emerging infectious zoonosis.

## Results

We provide numerical simulations to illustrate potential fates of a zoonotic epidemic in reservoir hosts, intermediate hosts, and humans. To provide a baseline for these simulations, we use parameters corresponding to highly pathogenic avian influenza (Table 5).

To elucidate the effects of the interspecies transmission parameters—$p_d$, $p_h$, and $\mu$—we simulate an outbreak of avian influenza mutating from a low-pathogenic to a highly-pathogenic strain in an intermediate host. One of the best-known examples of a zoonosis with an intermediate host, avian influenza spreads from wild birds to domestic poultry to humans, a process for which there is some publicly available data. Seeding the model with the parameters shown in Table 5 (and assuming that $\beta_w = \beta_d$, $\gamma_w = \gamma_d$), we obtain the result shown in Fig 2.

**Table 5. Parameter values and sources for the model. Due to a lack of data for transmission parameters in wild animals, we assume $\beta_w$, $\gamma_w$, $b_w$, and $m_w$ to be equivalent to their counterparts in domestic animals.** The timesteps are given in days.

| Parameter | Value | Source |
|---|---|---|
| initial $S_w$ | 0.5 | [35] |
| initial $I_w$ | 0.5 | [35] |
| $p_d$ | 0.51 | [35] |
| $\beta_d$ | 0.89 | [36] |
| $\gamma_d$ | 0.981 | [36] |
| $b_d$ | 1 | assumed |
| $m_d$ | 1 | assumed |
| $p_h$ | 0.207 | [37] |
| $\beta_h$ | 0.078 | [37] |
| $\gamma_h$ | 0.091 | [37] |
| $b_h$ | 0.0118 | CDC |
| $m_h$ | 0.009 | CDC |
| $\mu$ | 0.499 | [35] |

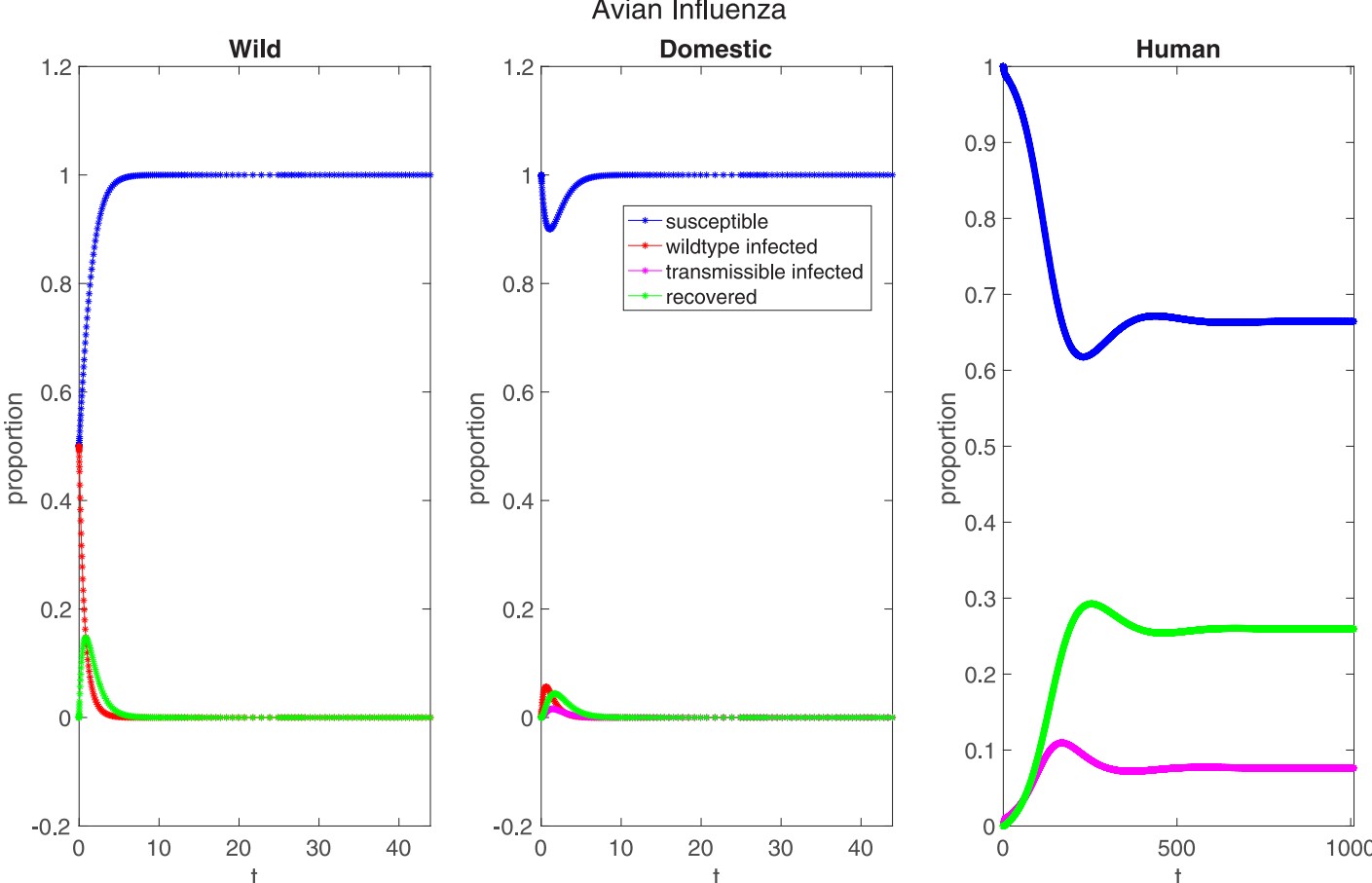

**Fig 2. A simulation of low-pathogenic avian influenza mutating to high-pathogenic avian influenza.** Parameters are as shown in Table 5. While the epidemic dies out in the animal species, its $R_0$ is 2.0871, allowing an epidemic to persist in humans.

This example–which uses the most data publicly available–shows that even if a pathogen's $R_0$ is less than one in both wild and intermediate hosts, it can still establish itself in the human population. Here, both strains of avian influenza fade in the animal populations while establishing an endemic equilibrium in the human population, with a maximum of 10.94% and an equilibrium of 7.65% of the population infected over a time span an order of magnitude larger than that necessary in the previous examples ($t$ = 2000 days, not shown in the figure). Although the particular numbers will change with more exact disease parameters, these simulations illustrate that with nonzero transmission parameters, an initial infection in an upstream host species will spread to an endemic equilibrium in downstream ones even if the pathogen fails to establish itself in its animal hosts. This result indicates that human epidemics can occur even without correspondingly severe outbreaks in animals.

We further evaluate the effect of varying the interspecies transmission parameters $p_d$, $\mu$, and $p_h$ on the equilibrium values $I_d^*$, $T_d^*$, and $I_h^*$ after 3000 days, in addition to $\beta_d$ and $\beta_h$ for comparison. To produce the graphs in Fig 3, we vary the parameter in question from 0.01 to 5 (since values of 0 inevitably lead to a disease-free equilibrium in the human compartment), with a step size of 0.1, holding the other values constant at the endemic equilibrium parameters detailed above.

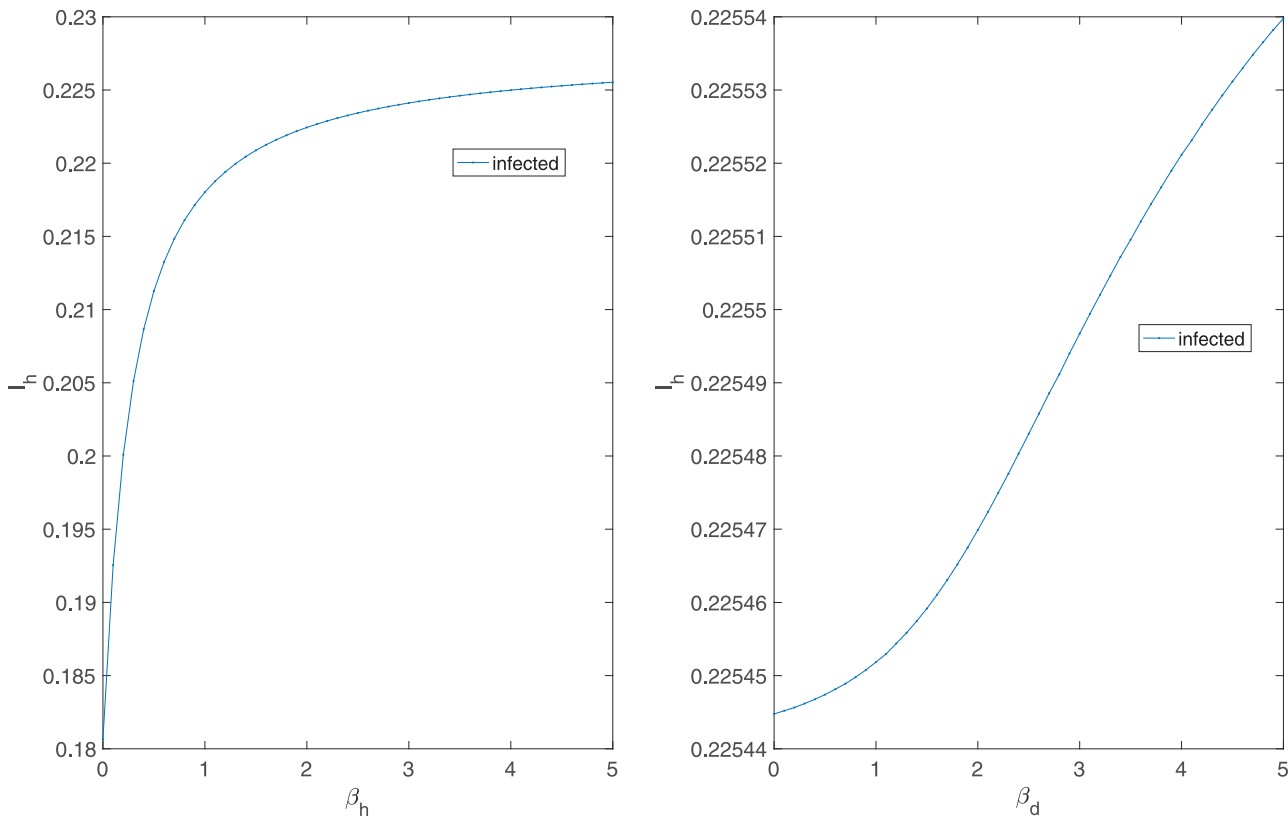

**Fig 3. $\beta_h$ (right) and $\beta_d$ (left) are directly proportional to the proportion of humans infected with the mutated strain.**

Similarly, we vary $p_d$, $\mu$, and $p_h$ to examine the effect of these parameters on the proportion of infected humans, finding that while increasing the mutation and intermediate host-human contact rate increases this proportion, increasing $p_d$ lowers it, as a larger contact rate between wild and domestic animals leads to a larger proportion of animals infected with the non-human-transmissible strain and thus unable to pass the disease to humans. Fig 4 shows heat-maps relating the interspecies transmission rates to the proportion of humans infected for four different values of $\mu$. This result is robust even for a pathogen that cannot spread among humans; as shown in Fig 5, even decreasing $\beta_h$ to 0 still leads to an endemic equilibrium, with $I_h^* > 0$.

The importance of the interspecies transmission parameters is reflected in Fig 5, which show that even when the transmission rates of the pathogen in humans or domestic animals is set to 0, the disease can reach an endemic equilibrium in humans. Further, only by setting one or more of the interspecies transmission parameters $\mu$, $p_d$, $p_h$ to 0 can the model avoid an endemic equilibrium in humans. In particular, the pathogen can persist in humans even if $\beta_h$ = 0. The results of these numerical simulations show that varying $p_d$ and $\mu$ can change the relative prevalence of domestic animals infected with the wildtype and human-transmissible strains, which in turn can change the proportion of infected humans. Thus, the interspecies transmission parameters should be primary targets for intervention to lower the proportion of infected humans in this model.

While varying traditional epidemic parameters such as $\beta_i$ and $\gamma_i$ can change the relative numbers of individuals in each compartment, we show that only $p_d$, $p_h$, and $\mu$ control the movement of a zoonotic epidemic between species, a result detailed by the simulations above.

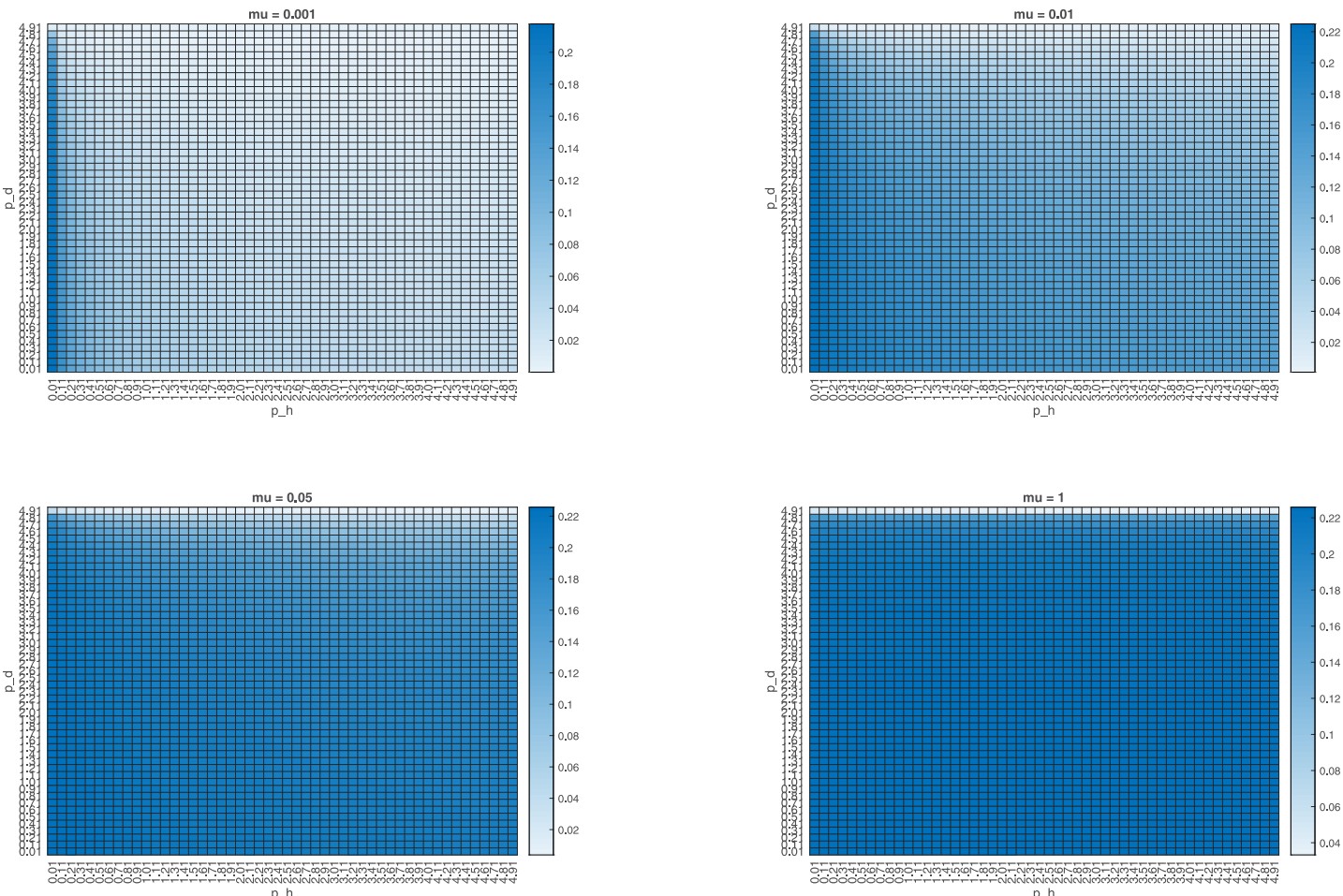

**Fig 4. Graphing the equilibrium proportion of infected humans ($I_h$) against $p_h$ and $p_d$ for four different values of $\mu$, with $\beta_h = 0.078$.** Parameters are as in Table 5, with $\beta_w = \beta_d = 0.118^*5$. While intracompartmental reproductive numbers vary between simulations, $R_0$ for all four simulations is 2.2463.

These results show that a zoonotic pathogen can establish itself in the human population as long as it is seeded with an initial infection in the wild compartment and $p_d$, $p_h$ and $\mu$ are nonzero, even if the human-transmissible strain is incapable of being transmitted between humans.

## Discussion

Since the spillover potential of the pathogen depends on $p_d$, $p_h$, and $\mu$, we distinguish between intracompartment parameters–the transmission and recovery rates $\beta_i$ and $\gamma_i$, which describe interactions in a single species–and intercompartment parameters–the spillover and mutation probabilities $p_d$, $p_h$, and $\mu$–which govern interactions between members of two species. We illustrate through several numerical examples that the intercompartmental parameters, and the initial proportion of infected wild animals, have the potential to alter the global dynamics of the three-species system to a disease-free equilibrium. Changing intracompartmental para-menters only changes the relative proportions of each type of individual present at an equilib-rium, not the stability of the equilibria, while modifying the values of intercompartmental parameters can change the global behavior of the pathogen. Isolating these parameters thus provides suggestions for possible interventions. In particular, while many parameters of the

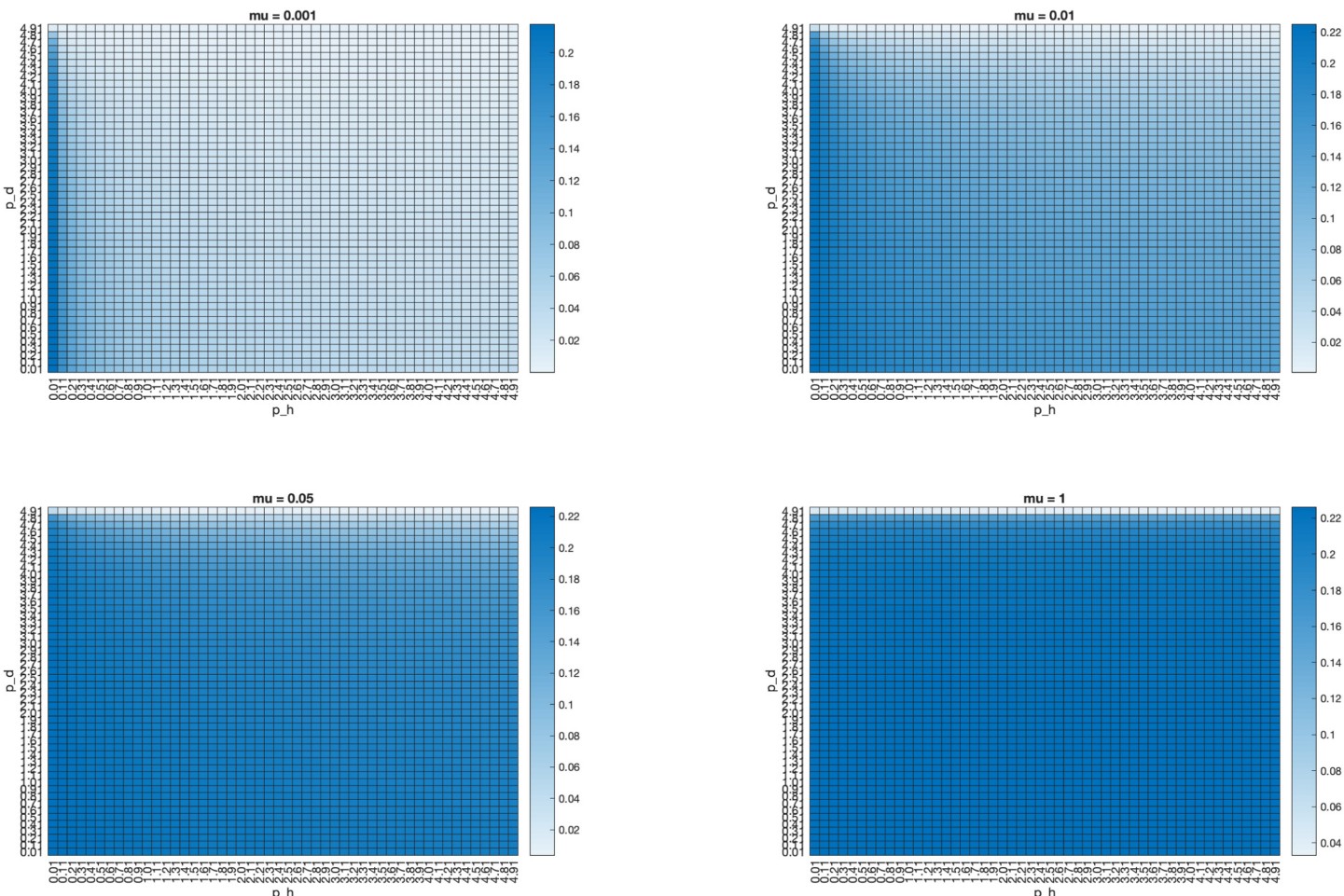

**Fig 5. Graphing the equilibrium proportion of infected humans ($I_h$) against $p_h$ and $p_d$ for four different values of $\mu$, with $\beta_h = 0$.** Parameters are as in Table 5, with $\beta_w = \beta_d = 0.118^*5$. While intracompartmental reproductive numbers vary between simulations, $R_0$ for all four simulations is 2.2463.

model can be changed by human interventions, the only effective route for eliminating the possibility of a zoonotic epidemic in humans is to eliminate contact between species or the possibility of pathogen mutation, an impossible requirement in any real system.

Reflecting the lack of data for zoonoses over their entire range of species, the sources used for the parameters in Table 5 reflect different strains of avian influenza. While the variety and inconsistency of these sources reflects the need for more data and research into the actual effects of particular zoonoses [35–38]), and it is crucial for public health interventions based on a mathematical model to know the accuracy of each parameter, their specific values are relatively unimportant for the theoretical results presented here, as the analysis of the system holds for all parameter values. The lack of large, publicly available data sets, especially regarding the prevalence of zoonotic infections in wild and domestic animals and the values for $p_d$, $p_h$, and $\mu$, limits our ability to refine any model [5–7], and so gathering such data should form a key component of future efforts.

This complete simulation of an emerging zoonosis shows that even in cases where the disease dies out in the wild compartment and would fail without an external force of infection in the domestic one, it can establish an endemic equilibrium in humans. Further, this result holds even if $\beta_h = 0$, reflecting a pathogen in Stage 1 of the traditional categorization for zoonoses

that would not be deemed a pandemic threat under that framework and suggesting that the threat posed by zoonoses is more severe than previously assumed. This result indicates that even the slightest possibility of contact between species or selection for a pathogen more suited to humans raises $p_d$, $p_h$, or $\mu$ above 0 and thus can lead to an endemic infection in humans. While these factors may be negligible in real populations, our results that the threat of an emerging zoonosis cannot be completely erased even with extraordinarily effective interventions mathematically confirm the focus on prioritizing zoonoses and offer a warning for public health officials.

This paper introduces a model capable of replicating all stages of the emergence of a zoonosis with an intermediate host; given adequate data, future research could adapt this model to any specific emerging zoonosis. To keep this work at a preliminary level and to maximize its use in more specialized contexts, we have not considered further modifications to the SIR prototype model such as loss of immunity (SIRS) or exposure time (SEIR), or possible variation patterns in the number of infected reservoir hosts, such as seasonal migration. In particular, the model incorporates neither pathogen virulence in new host species nor logistic growth limits on populations. It is thus best suited to a pathogen that does not cause significant host mortality, and future research provides an excellent opportunity to investigate the complexities arising in more virulent diseases. Future models could also incorporate backwards transmission to wild animals, direct interactions between humans and wild reservoirs, and interactions between different pathogens in an intermediate host [6]. The effect of different transmission rates for the two strains circulating in the intermediate host, as well as the relaxation of the assumption of mass action in the human compartment, also provide areas for future study. Finally, we were unable to investigate disease dynamics in individual hosts, with little data regarding the effect of different expressions of pathogen genotypes or animal superspreaders on transmissibility in humans [6]. As this effect is abstracted by our parameter $\mu$, delving deeper into individual-host pathogen dynamics such as cellular entry and replication [7] has the potential to improve our model. No emerging infected disease has been predicted before infecting humans [28], although progress is being made on identifying disease 'hotspots' [39], and this inability reinforces the importance of studying the factors that lead to successful spillover and define transmission rates between species [28].

This research suggests future avenues of exploration for both researchers and policymakers seeking to understand and control the spread of an emerging infectious zoonosis, and proves that interspecies connections are critical to controlling and understanding the effect of an emerging zoonosis on human populations. We show that with nonzero transmission parameters and an initial population of infected wild animals, a pathogen can fail to achieve traditional markers of success, such as stage 3 transmissibility, and still maintain an endemic equilibrium in the human population. This concerning result for public health offers areas in which policy rather than medical interventions can be more effective in controlling disease.

## Conclusion

We establish that this model of the entire path of an emerging infectious zoonosis has one unique disease-free equilibrium and one endemic equilibrium, and that the stability of these points depends on $p_d$, $p_h$, and $\mu$, the contact probabilities between species and the pathogen's rate of mutation. Accurately identifying and describing the dynamics of a pathogen circulating in wild and domestic animals provides an invaluable opportunity to avoid risk to humans [28], and can be used to guide public health interventions for emerging zoonotic diseases.

With the ability to study the emergence of a zoonosis with an intermediate host, first quantified by the model introduced here, scientists and policymakers have a more refined tool with

which to study and confront the emergence of a new pandemic into the human population. To our knowledge, this is the first model that accounts for the entire course–from infected wild animals, through mutation in an intermediate host, to an endemic equilibrium in humans–of the type of zoonotic pathogen the World Health Organization ranks in the highest tier of priorities for research and development, and so provides a significant step forward in its study.

Our results primarily offer a warning to public health officials: without drastic interventions to lower interspecies interactions or pathogen mutation rates, zoonoses with the capacity to mutate in a human-adjacent intermediate host can spread to humans even if they are not viable in a human population alone. More fundamentally to the field of mathematical epidemiology, this result confirms previously held beliefs–unquantified until now–about the philosophical importance of zoonoses to humanity. It is a pillar of the movement variously called "global", "one", or "planetary health" that human populations cannot isolate themselves from changes that affect other species with interventions targeting only humans. By mathematically linking the progress of a zoonotic epidemic to parameters governing interactions between species, this model shows that the framework of an interconnected human and natural world that implicitly underlies much of the analysis in this field in the last twenty years agrees with the mathematics of infectious disease, quantifying and confirming a widespread belief in global health.

## Supporting information

**S1 Appendix.**
(PDF)

**S1 File.**
(ZIP)

## Author Contributions

**Conceptualization:** Katherine Royce, Feng Fu.

**Investigation:** Katherine Royce, Feng Fu.

**Methodology:** Katherine Royce, Feng Fu.

**Writing – original draft:** Katherine Royce, Feng Fu.

**Writing – review & editing:** Katherine Royce, Feng Fu.

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
