## [Decision Letter · Decision Letter 0]

22 Jan 2020

PONE-D-19-34625

Mathematically modeling spillovers of an emerging infectious zoonosis with an intermediate host

PLOS ONE

Dear Ms Royce,

Thank you for submitting your manuscript to PLOS ONE. After careful consideration, we feel that it has merit but does not fully meet PLOS ONE’s publication criteria as it currently stands. Therefore, we invite you to submit a revised version of the manuscript that addresses the points raised by the two reviewers during the review process. 

We would appreciate receiving your revised manuscript by Mar 07 2020 11:59PM. To enhance the reproducibility of your results, we recommend that if applicable you deposit your laboratory protocols in protocols.io, where a protocol can be assigned its own identifier (DOI) such that it can be cited independently in the future. For instructions see: http://journals.plos.org/plosone/s/submission-guidelines#loc-laboratory-protocols

We look forward to receiving your revised manuscript.

Kind regards,

Chris T. Bauch, Ph.D.

Academic Editor

PLOS ONE

Journal Requirements:

Reviewers' comments:

Reviewer's Responses to Questions

**Comments to the Author**

1. Is the manuscript technically sound, and do the data support the conclusions?

Reviewer #1: Partly

Reviewer #2: No

2. Has the statistical analysis been performed appropriately and rigorously? 

Reviewer #1: Yes

Reviewer #2: No

3. Have the authors made all data underlying the findings in their manuscript fully available?

Reviewer #1: No

Reviewer #2: Yes

4. Is the manuscript presented in an intelligible fashion and written in standard English?

Reviewer #1: Yes

Reviewer #2: No

5. Review Comments to the Author

Reviewer #1: The authors here develop a simple ODE model of pathogen spillover from a wildlife reservoir to an intermediate host to a human. The novelty of the approach lies in considering mutation within the intermediate host and the effects this has on pathogen invasion and equilibrium prevalence. The model is generally sound and the analytic results are robust. My primary comments concern better framing the study within the recent literature, the sensitivity of the model to some alternative structure, and more generally restricting the paper to the key results and information. These comments are explained below.

1) The authors frame this paper around the lack of a mathematical framework for zoonotic spillover that considers the processes occurring from the reservoir to the recipient (human) host (e.g., L19, L33). Unfortunately, this ignores some foundational work over the past few years that have developed similar (but different) modeling approaches. For example, Plowright et al. 2017 traces spillover from reservoir disease dynamics to establishment in a recipient host, Childs et al. 2019 traces the dynamics of yellow fever from reservoirs to human epidemics, Washburne et al. 2019 develop a percolation model of spillover from donor to recipient hosts, and Faust et al. 20177 develop an explicit mathematical model of spillover from wildlife to humans in the context of land conversion. I think the authors should review the recent literature more thoroughly and identify the novel gap their study addresses. It is fine to say that although recent modeling efforts have addressed the spillover process from reservoir to recipient hosts, the role of intermediate hosts as amplifiers or mutators of a pathogen (a defining part of zoonotic spillover) remains underdeveloped and lacks a strong theoretical foundation.

2) The authors have understandably kept their model relatively simple to facilitate generalizable insights (e.g., as described in L593). However, I do have a concern about two components of the model, which are independent of the general infection process. First, the authors assume a constant birth rate b per each host species, but density-dependent birth (i.e., logistic growth) seems far more appropriate than exponential growth. I suggest the authors use logistic growth of each host instead (e.g., see Faust et al. 2017). Second, the authors assume that the pathogen is not virulent (no disease-induced mortality) in either the wild host, intermediate host, or human host, which does not seem to reflect the actual context of many zoonoses (e.g., mortality from AIV in poultry and humans, mortality from Nipah virus in pigs and humans, mortality of horses and humans from Hendra virus). Are your results sensitive to including disease-induced mortality alongside the baseline mortality per host?

3) The authors also conduct a local sensitivity analysis, in which pd, ph, and mu are varied to observe effects on the equilibria infection prevalences per host. Although the local analysis seems fine (i.e., holding all other parameters constant), I think the results could be made more robust by systematically varying all combinations of pd and ph for a few values of mu (e.g., heat maps with pd and ph as the axes, prevalence as the surface, for 3-4 different values of mu to represent different degrees of mutation). This could better reflect where in parameter space human prevalence is maximized. Alternatively, the authors could use a Latin hypercube sampling approach to vary all these parameters simultaneously.

4) As a more editorial note, I think the manuscript would benefit from a bit more selectivity in text and results. In particular, I found the Introduction (L1-159) to be rather long, and much of the information could be condensed into several clear paragraphs (e.g., introduce zoonoses and spillover, discuss the role of intermediate hosts, highlight previous work on mutation, describe the aims of the model). I also think the manuscript would be easier to follow if the authors adopted the more traditional manuscript format of Introduction, Methods, Results, Discussion. Lastly, given that there are 13 figures in the manuscript, I suggest the authors consider what are the key results to show the reader, include those key figures (or combine figures into multi-panel figures), and move other results to the supplement. This would help streamline the MS.

5) As a minor point, the authors often use “zoonotic disease” when I think they mean “zoonotic pathogen”. Pathogens are what is being transmitted between hosts (or mutated within a host), rather than the disease manifestation.

6) In several parts of the methods, the authors include some additional description of the methods used that, at times, distract from the main text. For example, the discussion of dengue (L174-179) seems out of place here and distracts from the methods being discussed. Similarly, the discussion of different applications of the next-generation matrix technique (L248-252) distract from the main methods. It’s OK to simply state that you used the next-generation matrix method to derive R0 and leave it at that.

7) I’m not sure Table 5 and 6 are necessary, given the corresponding figures. The actual numerical results here will vary based on other parameter values (e.g., beta, b, mu), so the tables not very informative when we can just view the graphs.

8) What software did the authors use for visualizing analytic results and running numerical simulations? Note that a condition of publication in PLoS ONE is to make underlying data fully available without restriction. The authors should upload the code used to generate results (Matlab, R, etc) as a supplemental file.

Reviewer #2: My review is uploaded as an attachment.

A mathematical model for a zoonosis is studied and applied to avian influenza. The model consists of three different

populations, wild reservoir, intermediate domestic animals, and humans. The infection is spread from the wild to

intermediate to humans. Spread to humans only occurs after a mutation of the virus in the intermediate host. The

introduction is well written and the concept is interesting.

Unfortunately, the analysis is simplistic and incorrect. As written it is difficult to follow what is shown in the large number

of Figures 4-13 in the simulations. The reproductive numbers for each population are not provided. The model is closely

related to a multigroup model with two types of infection in the intermediate group.

The value of R_0 can be calculated at the disease-free equilibrium (dfe) by applying Theorem 2 in [37] as in lines 390-

395. In addition, if the conditions (A1) and (A5) hold in [37], then Theorem 2 can be applied to show that the dfe is

locally asymptotically stable. Theorems 1-5 and the lemmas are not required. The analyses for each population, Wild,

Domestic, and Human, are trivial and are not needed when considering the entire system. It is nontrivial to show that

the endemic equilibrium E_e is locally stable. This requires the Routh-Hurwitz criteria applied to the Jacobian matrix

evaluated at E_e for all 10 differential equations. Therefore, Theorem 6 is incorrect. Alternately given specific parameter

values the eigenvalues of the Jacobian matrix can be computed

In several figures, either the wild or domestic animals approach the dfe. Why? The term ``reservoir" implies that the

basic reproductive number beta_wb_w/(m_w(gamma_w+m_w))>1 (typo in (14)). Methods for parameter sensitivity,

such as Latin hypercube sampling and partial rank correlation coefficient, may be useful. The parameters p_d, \\mu, and

p_h are important, as they connect the three populations.

The differential equations can be written in the text, rather than in a Table. There are three reproductive numbers, one

for each population. A suggestion is to define R_0 in (14) as follows:

R_0=max{R_0^w, R_0^d, R_0^h}.

Then the three reproductive numbers can be given for the figures.

6. PLOS authors have the option to publish the peer review history of their article (what does this mean?). If published, this will include your full peer review and any attached files.

Reviewer #1: No

Reviewer #2: No

---

## [Author Response · Author response to Decision Letter 0]

12 Apr 2020

We have incorporated ideas from Plowright et al. 2017, Childs et al. 2019, and Washburne et al. 2019. Per Faust et al. 2017, we have included a suggestion to incorporate logistic growth of each host in future research, but our deterministic model is not sensitive to disease-induced mortality (since, as noted in the Appendix, any infection in an upstream host will spill over to downstream ones even if the epidemic dies out among that particular species). A stochastic model, which we suggest as another possible extension of our work, may retain the sensitivity to disease-induced mortality that this model lacks; however, this model was intended to provide a preliminary basis for quantitative investigation of zoonoses with intermediate hosts, and so the question of the effects of disease-induced mortality is beyond the scope of the current paper. We have incorporated Figures 4 and 5, heat maps with ph and pd as the axes and Ih as the surface, for four values of mu. We performed this experiment for beta_h = 0.078 and for beta_h = 0. We have shortened the introduction to 4 paragraphs and conformed to a more traditional manuscript format, following the suggested structure. Further, we have combined the original figures into 5 final versions, consolidating our findings. We have revised our description of zoonotic pathogens to match the behavior being discussed in each instance. We have deleted the discussions of dengue (L174-179) and of applications (L248-252), as well as Tables 5 and 6, and moved a mathematical proof of our results to an appendix. Finally, we have included the Matlab code used to produce our results as a supplemental file.

---

## [Editor Report · Decision Letter 1]

16 Apr 2020

PONE-D-19-34625R1

Mathematically modeling spillovers of an emerging infectious zoonosis with an intermediate host

PLOS ONE

Dear Ms Royce,

Thank you for submitting your manuscript to PLOS ONE. After careful consideration, we feel that it has merit but does not fully meet PLOS ONE’s publication criteria as it currently stands. Therefore, we invite you to submit a revised version of the manuscript that addresses the points raised during the review process.

We would appreciate receiving your revised manuscript by May 31 2020 11:59PM. To enhance the reproducibility of your results, we recommend that if applicable you deposit your laboratory protocols in protocols.io, where a protocol can be assigned its own identifier (DOI) such that it can be cited independently in the future. For instructions see: http://journals.plos.org/plosone/s/submission-guidelines#loc-laboratory-protocols

We look forward to receiving your revised manuscript.

Kind regards,

Chris T. Bauch, Ph.D.

Academic Editor

PLOS ONE

Additional Editor Comments (if provided):

Please resubmit this paper with a detailed, point-by-point response to the reviewer comments, as requested in the previous decision letter.

---

## [Author Response · Author response to Decision Letter 1]

26 May 2020

Journal Requirements: we have ensured that the figures are properly labeled and that the manuscript complies with the PLoS ONE Latex template. We have deleted the supplementary writing and included the relevant analysis in the Methods section, and have ensured that our supplementary code complies with formatting requirements.

Reviewer 1:We have incorporated ideas from Plowright et al. 2017, Childs et al. 2019, and Washburne et al. 2019, highlighting the differences between the conceptual models of spillover dynamics that do not include disease dynamics in nonhuman species and our model, which explicitly accounts for disease mutation in a nonhuman host. Per Faust et al. 2017, we have included a suggestion to incorporate logistic growth of each host in future research, but our deterministic model is not sensitive to disease-induced mortality. Our goal is to provide a conceptual framework for intermediate host transmission in many contexts, and we have thus kept the simpler assumption of constant birth and death rates in order to make the model accessible in many different contexts. A stochastic model, which we suggest as another possible extension of our work, may retain the sensitivity to disease-induced mortality that this model lacks; however, this model was intended to provide a preliminary basis for quantitative investigation of zoonoses with intermediate hosts, and so the question of the effects of disease-induced mortality is beyond the scope of the current paper. We have incorporated Figures 4 and 5, heat maps with p_h and p_d as the axes and I_h as the surface, for four values of mu, giving a more complete explanation of where in parameter space the proportion of infected humans is maximized. Further, we performed this experiment for two values of beta_h, distinguishing between a human-to-human transmissible pathogen and one that cannot be transmitted among humans. We have shortened the introduction to 4 paragraphs and conformed to a more traditional manuscript format, following the suggested structure. Further, we have combined the original figures into 5 final versions, consolidating our findings. We have revised our description of zoonotic pathogens to match the behavior being discussed in each instance. We have deleted the discussions of dengue (L174-179) and of applications (L248-252), and moved a mathematical proof of our results to an appendix. We have deleted Tables 5 and 6, replacing them with the heatmaps described above. We have included the Matlab code used to produce our results as a supplemental file.

Reviewer 2:We have included a formula for R0 in the model and noted the similarities to multigroup models, as well as including code to calculate R0 given specific parameter values. In addition, we have included a reference to the Routh-Hurwitz criteria to ensure completeness of the analysis. We have corrected the typo in (14) and clarified the importance of p_d, p_h, and mu. While we considered Latin hypercube sampling, in particular, the aim of this paper is to introduce a new type of model, with the goal of prompting further research, and we have thus kept the analysis as simple as possible. We have clarified (lines 161-181) the distinction between intracompartmental reproductive numbers and the global reproductive number, explaining why the pathogen can fade in animal species while reaching an endemic equilibrium in humans. We have also noted that this behavior may cause the wild species not to be a 'reservoir host' in the true sense of the term (line 176). While we have kept the differential equations in a table, following conventions in mathematical biology, we have explicitly stated R0 on line 160 and given a numerical R0 for each of the figures.

We thank the reviewers for their feedback.

---

## [Decision Letter · Decision Letter 2]

15 Jun 2020

PONE-D-19-34625R2

Mathematically modeling spillovers of an emerging infectious zoonosis with an intermediate host

PLOS ONE

Dear Dr. Royce,

Thank you for submitting your manuscript to PLOS ONE. After careful consideration, we feel that it has merit but does not fully meet PLOS ONE’s publication criteria as it currently stands. Therefore, we invite you to submit a revised version of the manuscript that addresses the points raised during the review process.

Please address the latest round of comments of the reviewers. Both reviewers appreciated that significant changes had been made, but felt that the manuscript would benefit significantly from a few more changes. Especially, reviewer #1 felt that the revised manuscript should be more clear that the model only applies to cases where there is no mortality from pathogen virulence, and that reviewer also noted that some requested changes from the first round of review had not been implemented yet.  

We look forward to receiving your revised manuscript.

Kind regards,

Chris T. Bauch, Ph.D.

Academic Editor

PLOS ONE

Reviewers' comments:

Reviewer's Responses to Questions

**Comments to the Author**

1. If the authors have adequately addressed your comments raised in a previous round of review and you feel that this manuscript is now acceptable for publication, you may indicate that here to bypass the “Comments to the Author” section, enter your conflict of interest statement in the “Confidential to Editor” section, and submit your "Accept" recommendation.

Reviewer #1: (No Response)

Reviewer #2: (No Response)

2. Is the manuscript technically sound, and do the data support the conclusions?

Reviewer #1: Partly

Reviewer #2: Yes

3. Has the statistical analysis been performed appropriately and rigorously? 

Reviewer #1: No

Reviewer #2: N/A

4. Have the authors made all data underlying the findings in their manuscript fully available?

Reviewer #1: Yes

Reviewer #2: Yes

5. Is the manuscript presented in an intelligible fashion and written in standard English?

Reviewer #1: Yes

Reviewer #2: No

6. Review Comments to the Author

Reviewer #1: I think the authors have done a generally nice job re-contextualizing their study in the recent literature, and the new sensitivity analyses are helpful to more thoroughly explore parameter space. However, I do have remaining concerns.

I still don't quite agree about ignoring logistic growth or pathogen virulence. Especially for the latter, many zoonotic pathogens are likely avirulent in the reservoir but virulent in the intermediate host and humans (e.g., Nipah virus, Ebola virus, etc). The authors note “Per Faust et al. 2017, we have included a suggestion to incorporate logistic growth of each host in future research, but our deterministic model is not sensitive to disease-induced mortality”, but I don’t see where this information was included in the revision. If the authors feel strongly that an additional mortality term should not be added, then I think they need to discuss the kinds of host-pathogen systems for which their assumptions (exponential growth, avirulence) are best met. For example, avian influenza causes high mortality in poultry (and of course humans; L101). This point could be added in the current section on model assumptions (L140-153).

L21: “in particular, there is no unifying mathematical theory or set of principles that can be used to frame discussions of zoonotic spillovers” Although the authors now better describe more recent theoretical work on pathogen spillover, I think statements such as these do not reflect the field and need to be toned down.

L46: Perhaps reword to “Table 1…shows notable case studies of zoonoses with domesticated species as intermediate hosts”

L75: This statement isn’t entirely accurate. Although SIR models of multi-host pathogens are often applied to vector-borne diseases, I wouldn’t say this captures the majority of such models.

Table 5: What are the units of time in the parameterization here?

Discussion: I was surprised to see no discussion of non-exponential growth or virulence here, given earlier comments.

Figure 2: The figure would be easier to read if the Y axis was consistent across panels (zero to 1, rather than -0.2 to 1.2 for some panels). Please label the X axis (e.g., “time in years”).

Figure 3: Labeling the axes in parameter names, in addition to the symbols, would be helpful (e.g., transmission rate among humans, Bh). Note that there is no need to add a key in this figure (“infected”), because that is implied by the Y axis. It would also be worth clarifying that altering betah has essentially no impact on Ih, given that Y ranges from 0.22544 to 0.22554. The authors might consider making the Y axes the same in both figures to drive this point home.

Figure 4 and 5: These figures are nice in concept but are unfortunately somewhat difficult to read. Simplifying the X and Y axes to only show 5-6 evenly spaced values would be helpful. The colorbars also need keys / titles.

Reviewer #2: Please see the attachment at the end of this letter. It contains a pdf file of all of my suggestions for revision.

7. PLOS authors have the option to publish the peer review history of their article (what does this mean?). If published, this will include your full peer review and any attached files.

Reviewer #1: No

Reviewer #2: No

---

## [Author Response · Author response to Decision Letter 2]

27 Jul 2020

We have clarified that the revised model applies only to diseases without mortality from pathogen virulence, and clarified Figures 2-5. In addition, we have implemented the revisions as described in the Response to Reviewers and edited the references to remove unused papers.

---

## [Editor Report · Decision Letter 3]

30 Jul 2020

PONE-D-19-34625R3

Mathematically modeling spillovers of an emerging infectious zoonosis with an intermediate host

PLOS ONE

Dear Dr. Royce,

Thank you for submitting your manuscript to PLOS ONE.  We feel that you have addressed the reviewer comments in a satisfactory manner, but there remain a few minor issues that should be addressed before it meets PLOS ONE publication criteria. Therefore, we invite you to submit a revised version of the manuscript that addresses the points.

1) lines 166-168: "our model is thus best suited to the first phase of diseases such as Covid-19 or Ebola, which spread between hosts in days but can take weeks to kill." -- even COVID-19 and ebola can kill hosts very quickly, in a matter of days, so I don't think this is the best example to use.  Instead, the authors should mention the example of 2009 H1N1 pandemic influenza, which was remarkably avirulent.  The authors should also revise line 329 accordingly. 

2) line 327: " In particular, as noted during revisions, the model incorporates neither pathogen virulence in new host species nor logistic growth limits on 3populations-- the authors should remove the reference to paper revisions, since papers do not normally make reference to their own review process. 

We look forward to receiving your revised manuscript.

Kind regards,

Chris T. Bauch, Ph.D.

Academic Editor

PLOS ONE

---

## [Author Response · Author response to Decision Letter 3]

31 Jul 2020

I have implemented the two changes suggested (replacing a reference to Covid-19 and Ebola with 2009 pandemic influenza as an example of a zoonosis with notable avirulence in new host species and removing a reference to the revision process), in addition to revising Table 1 so that the sources are cited in numerical order.

---

## [Editor Report · Decision Letter 4]

4 Aug 2020

Mathematically modeling spillovers of an emerging infectious zoonosis with an intermediate host

PONE-D-19-34625R4

Dear Dr. Royce,

We’re pleased to inform you that your manuscript has been judged scientifically suitable for publication and will be formally accepted for publication once it meets all outstanding technical requirements.

Kind regards,

Chris T. Bauch, Ph.D.

Academic Editor

PLOS ONE
---

## [Editor Report · Acceptance letter]

7 Aug 2020

PONE-D-19-34625R4 

Mathematically modeling spillovers of an emerging infectious zoonosis with an intermediate host 

Dear Dr. Royce:

I'm pleased to inform you that your manuscript has been deemed suitable for publication in PLOS ONE. Congratulations! Your manuscript is now with our production department. 

Kind regards, 

on behalf of

Professor Chris T. Bauch 

Academic Editor

PLOS ONE